# Application of Artificial Intelligence Methods for Imaging of Spinal Metastasis

**DOI:** 10.3390/cancers14164025

**Published:** 2022-08-20

**Authors:** Wilson Ong, Lei Zhu, Wenqiao Zhang, Tricia Kuah, Desmond Shi Wei Lim, Xi Zhen Low, Yee Liang Thian, Ee Chin Teo, Jiong Hao Tan, Naresh Kumar, Balamurugan A. Vellayappan, Beng Chin Ooi, Swee Tian Quek, Andrew Makmur, James Thomas Patrick Decourcy Hallinan

**Affiliations:** 1Department of Diagnostic Imaging, National University Hospital, 5 Lower Kent Ridge Rd., Singapore 119074, Singapore; 2Department of Computer Science, School of Computing, National University of Singapore, 13 Computing Drive, Singapore 117417, Singapore; 3Department of Diagnostic Radiology, Yong Loo Lin School of Medicine, National University of Singapore, 10 Medical Drive, Singapore 117597, Singapore; 4University Spine Centre, Department of Orthopaedic Surgery, National University Health System, 1E, Lower Kent Ridge Road, Singapore 119228, Singapore; 5Department of Radiation Oncology, National University Cancer Institute Singapore, National University Hospital, Singapore 119074, Singapore

**Keywords:** artificial intelligence, machine learning, deep learning, spinal metastasis, imaging, applications

## Abstract

**Simple Summary:**

Spinal metastasis is the most common malignant disease of the spine, and its early diagnosis and treatment is important to prevent complications and improve quality of life. With the recent advances in medical imaging and artificial intelligence (AI), there is a dramatic rise in research related to computer-aided interpretation of spinal metastasis imaging. This study will review the current evidence for AI methods in spinal metastasis imaging using a systemic approach. Potential clinical applications of AI, designed to solve the issues frequently faced in the management of spinal metastasis, will also be discussed.

**Abstract:**

Spinal metastasis is the most common malignant disease of the spine. Recently, major advances in machine learning and artificial intelligence technology have led to their increased use in oncological imaging. The purpose of this study is to review and summarise the present evidence for artificial intelligence applications in the detection, classification and management of spinal metastasis, along with their potential integration into clinical practice. A systematic, detailed search of the main electronic medical databases was undertaken in concordance with the PRISMA guidelines. A total of 30 articles were retrieved from the database and reviewed. Key findings of current AI applications were compiled and summarised. The main clinical applications of AI techniques include image processing, diagnosis, decision support, treatment assistance and prognostic outcomes. In the realm of spinal oncology, artificial intelligence technologies have achieved relatively good performance and hold immense potential to aid clinicians, including enhancing work efficiency and reducing adverse events. Further research is required to validate the clinical performance of the AI tools and facilitate their integration into routine clinical practice.

## 1. Introduction

Spinal metastasis is a malignant process along the spine that is up to 35 times more common than any other primary malignant disease along the spine [1] and represents the third most common location for metastases [2]. Spinal metastasis can tremendously impact quality of life, secondary to complications such as pain due to fractures, spinal cord compression, neurological deficits [3,4], reduced mobility, bone marrow aplasia and hypercalcemia leading to symptoms such as constipation, polyuria, polydipsia, fatigue and even cardiac arrythmias and acute renal failure [5,6]. Therefore, the timely detection, diagnosis and optimal treatment of spinal metastases is essential to reduce complications and to improve patients’ quality of life [7].

Radiological investigations play a central role in the diagnosis and treatment planning of spinal metastases. Plain radiographs are a quick and inexpensive first-line investigation, although advanced modalities such as computed tomography (CT), magnetic resonance imaging (MRI), positron emission tomography (PET) and bone scintigraphy are all superior for the detection and classification of spinal metastases [8]. Different imaging modalities have their own advantages over each other in the assessment of spinal metastasis. CT with sensitivity and specificity of 79.2% and 92.3%, respectively, for the detection of spinal metastases [9], can be used to guide interventional procedures and also provides systemic staging [10]. Compared to CT, MRI has higher sensitivity and specificity of 94.1% and 94.2%, respectively, for spinal metastasis detection [9], and is radiation-free. MRI is the modality of choice for assessing metastatic spread to the bone marrow and associated epidural soft tissue extension [11,12]. 18F FDG-PET (flurodeoxyglucose) has sensitivity and specificity of 89.8% and 63.3%, respectively, although sensitivity varies among different histologies due to their innate metabolic activity [9,13]. In bone scintigraphy, the sensitivity and specificity are 80.0% and 92.8%, respectively, and it is the most widely available technique for the study of bone metastatic disease [8,9].

Recently, preliminary Artificial Intelligence (AI) techniques have demonstrated remarkable progress in medical imaging applications, especially in the field of oncology [14]. The two most popular machine learning techniques are radiomics-based feature analysis, along with convolutional neural networks (CNN). Radiomics-based techniques require extraction of several handcrafted features, which are then selected to provide a training set for deep learning-based image classification [15]. One drawback of the technique is that the selected handcrafted features remain limited to the knowledge of the radiologist or clinician, which could reduce the accuracy of the developed algorithm [16]. Machine learning along with deep learning techniques can directly learn important imaging features for classification without the need for handcrafted feature selection. This typically involves convolutional neural networks, and these techniques have been shown in the literature to have improved prediction accuracy for lesion detection, segmentation and treatment response in oncological imaging [17,18,19].

The use of AI techniques in various oncological imaging [20] for primary malignancies, including breast [21,22,23], renal [24,25], brain [26,27,28,29] and liver cancers [30,31,32,33] have been studied, with the majority showing exceptional prediction outcome, although few have been validated in a real clinical setting. Recently, there has been research into computer aided interpretation, radiomics and machine learning to optimise the treatment decisions for spinal metastasis using multimodal imaging [34]. This review article aims to provide an overview of the most clinically pertinent applications of machine learning (including radiomics and CNNs) in spinal oncology imaging.

## 2. Materials and Methods

### 2.1. Literature Search Strategy

A systematic, detailed literature search of the main electronic medical databases was undertaken in concordance with the PRISMA guidelines [35,36]. Electronic databases searched included the Web of Science, clinicaltrials.gov, MEDLINE, and PubMed (last date of the literature search was the 31 May 2022). Databases were searched for the following terms: (“spinal” OR “vertebral”) AND (“metastasis” OR “metastases”) AND (“radiomics”, OR “machine learning”, OR “deep learning”, OR “artificial intelligence”).

### 2.2. Study Screening and Selection Criteria

No limitations were stipulated for the literature search. The main inclusion criteria were research studies utilising radiomics techniques or deep learning to analyse spinal metastases. Other inclusion criteria included the following: (a) imaging analysis involving nuclear medicine studies, CT and/or MRI scans; (b) studies addressing the capacity to predict, diagnose and integrate the deep learning into clinical practice; (c) studies involving humans only; and (d) publications in the English language. Articles excluded from further analysis included case reports, conference abstracts, review articles, and editorial correspondence (e.g., letters, commentaries and opinion pieces). Duplicate publications, articles more than 20 years old and publications focusing on non-imaging radiomics techniques (e.g., isolated histopathology features) were also excluded from analysis. The literature search was finalised by reviewing the selected publication bibliographies to identify any relevant articles.

### 2.3. Data Extraction and Reporting

All selected articles were retrieved and tabulated into a Microsoft Excel spreadsheet (Microsoft Corporation, Washington, DC, USA). Details gathered from each of the selected articles were as follows:Article details: Authorship, Date of online or journal publication;Potential clinical utility: Tumour classification or segmentation, treatment or prognosis prediction;Patient population: Patients with suspected or known spinal metastases, with benign vertebral lesions which have undergone imaging evaluation;Study details: Study type, sample size (number of patients, data sets), type of imaging (CT, MR, bone scan or PET-CT), treatment information and outcome measures.

## 3. Results

### 3.1. Literature Search Results

The preliminary search through the main electronic medical databases (Figure 1) identified 76 relevant articles, which were screened using the aforementioned criteria. This screening culminated in the initial exclusion of 41 publications, and the residual 35 publications underwent further full-text analysis to determine inclusion. Upon detailed analysis of the text, a further 11 articles were removed as they were focused on cancer sites outside the spine, or had no application using imaging data including focused molecular and/or genomic studies. An additional six articles were included after manually reviewing the bibliography of the selected articles. Overall, this culminated in a total of 30 publications (Table 1) for in-depth analysis. Key findings of the current AI applications were compiled and summarised in this review.

Our search found 15/30 (50.0%) studies were MRI-based, 10/30 (33.3%) were CT-based, 3/30 (10.0%) were related to nuclear studies (PET-CT/SPECT), 1/30 (3.3%) were DEXA-based and 1/30 (3.3%) used MRI and CT evaluation. In terms of the techniques used, 8/30 (26.7%) studies used radiomics to differentiate spinal metastases from other pathological conditions, 12/30 (40.0%) studies focused on machine learning to detect spinal metastases and 2/30 (6.7%) studies used radiomics to predict pain response and risk of compression fractures from treated spinal metastases. In addition, 9/30 (30.0%) studies used radiomics to generate clinically useful parameters, of which 3/30 (10.0%) used machine learning to assist in vertebral metastasis segmentation and treatment planning, 4/30 studies (13.3%) used radiomics to determine the spinal tumour characteristics and likely primary malignancy and 2/30 (6.7%) studies focused on using deep learning to classify the complications of spinal metastases.

### 3.2. Artificial Intelligence (AI)

Artificial intelligence (AI) is a term referring to a machine’s computational ability to perform tasks that are comparable to those executed by humans. This is done by utilising unique inputs and then generating outputs with high added value [37]. With recent advances in medical imaging and ever increasing large amounts of digital image and report data, worldwide interest in AI for medical imaging continues to increase [38]. The rationale of using AI and computer-aided diagnostic (CAD) systems was initially thought to assist clinicians or radiologists in the detection of tumours or lesions which in turn increases efficiency, improves detection and reduces error rates [39]. As a result, efforts are ongoing to enhance the diagnostic ability of AI, and enhance its efficiency so that is can be successfully translated into clinical practice [40]. With the advent of artificial neural networks, which are a class of architectures loosely based on how the human brain works [41], several computational learning models (mainly machine learning (ML) and deep learning (DL) algorithms) have been introduced and are largely responsible for the growth of AI in radiology. In general, the clinical applications of AI (Figure 2) can be broadly characterised into three categories for oncology imaging workflow: (1) detection of abnormalities; (2) characterisation of abnormalities, which includes image processing steps such as segmentation, differentiation and classification; and (3) integrated diagnostics, which include decision support for treatment decision and planning, treatment response and prognosis prediction.

### 3.3. Machine Learning (ML)

Machine learning is a field of AI in which models are trained for prediction using known datasets, from which the machine “learns”. The developed model then applies its knowledge to perform diagnostic tasks in unknown datasets [42]. The application of ML requires collection of data inputs that have been labelled by human experts (typically radiologists) or by direct extraction of the data using several different computational methods including supervised and unsupervised learning. Supervised machine learning models rely on labelled input data to learn the relationship with output training data [43], and are often used to classify data or make predictions. On the other hand, unsupervised machine learning models learn from unlabelled raw training data to learn the relationships and patterns within the dataset and discover inherent trends within the data set [44,45]. Unsupervised models are mainly used as an efficient representation of the initial dataset (e.g., densities, distances or clustering through dataset statistical properties) and to better understand relationships or patterns within the datasets [46,47]. Such new representation can be an initial step prior to training a supervised model (e.g., identifying anomalies and outliers within the datasets), which could improve performance in the supervised model [48,49,50].

### 3.4. Deep Learning (DL)

Deep learning represents a subdivision of machine learning (Figure 3), and is modelled on the neuronal architecture within the brain. The technique leverages artificial neural networks, which involve several layers to solve complex medical imaging challenges [51]. The multiple layered structure enables the deep learning model or algorithm to actively learn knowledge from the imaging datasets and make predictions on unseen imaging data [52]. These deep learning techniques can provide accurate image classification (disease present/absent, or severity of disease), segmentation (pixel-based), and detection capability [53]

Deep learning allows for the ability to process and detect fundamental diagnostic patterns and features beyond that of human abilities. This has created a new field of deep learning termed Radiomics, in which imaging features related to important pathological and histological subtyping of tumours can be identified and used for the detection, differentiation and prognosis of unknown lesions [54].

### 3.5. Radiomics

Radiomics is a relatively new branch of machine learning that involves converting medical images containing important information related to tumour features into measurable and quantifiable data [55]. This information can then aid clinicians in the assessment of tumours by providing additional data about tumour behaviour and pathophysiology beyond that of current subjective visual interpretation (inferable by human eyes) [56,57], such as tumour subtyping and grading [58]. Combined with clinical and qualitative imaging data, radiomics has been shown to guide and improve medical decision making [59], and can be used to aid disease prediction, provide prognostic information, along with treatment response assessment [58]. In general, the workflow for deriving a radiomics model can be divided into several steps (Figure 4): data selection (input), radiological imaging evaluation and segmentation, image feature extraction in the regions of interest (ROIs) and exploratory analysis followed by modelling [55]. Depending on the type of imaging modality, the acquisition, technical specifications, software, segmentation of the ROIs, image feature extraction and structure of the predictive algorithm are all different and subject to several factors [60]. Machine learning methods including random decision forest, an ensemble learning method for classifying data using decision trees, can then be performed to validate and further evaluate the classification accuracy of the set of predictors [61]. These can then be applied in a clinical setting to potentially improve the diagnostic accuracy and prediction of survival post-treatment [62,63].

There are two key radiomics techniques, namely handcrafted-feature based and deep learning-based analysis [64]. Firstly, handcrafted-feature radiomics involves extraction of features from an area of interest (typically segmented). These features can be placed into groups based on shape [65], histogram criteria (first order statistics) [66], textural-based criteria (second-order statistics) [39] and other higher order statistical criteria [67]. Following this step, machine learning models can be developed to provide clinical predictions, including survival/prognostic information based on the handcrafted-features [62,68,69]. The models are also assessed on validation datasets to review their efficiency and sensitivity.

In contrast to handcrafted-feature radiomics, deep learning techniques rely on convolutional neural networks (CNN) or other architectures [70] to identify the most pertinent radiological imaging features without relying on prior feature descriptions. CNNs provide automated extraction of the most important features from the radiological imaging data using a cascading process, which can then be used for pattern recognition and training [71]. The generated dominant imaging features can undergo further processing, or exit the neural network and be used for machine learning model generation using algorithms similar to the feature-based radiomics method before validation. The main drawback of deep learning-based radiomics is the requirement for much larger training datasets, since feature extraction is required as part of the initial process compared to feature-based radiomics where the features are manually selected for analysis [72]. With recent advances in AI, this limitation can be circumvented through transfer learning, which is a technique that uses neural networks that were pre-trained for another separate but closely related purpose [73]. As such, by leveraging on the network’s prior knowledge, transfer learning reduces computational demand and the amount of training data required, but can still produce reliable performance.

Radiomics techniques have transformed the outlook of quantitative medical imaging research. Radiomics could provide rapid, comprehensive characterisation of tumours at minimal cost, which would act as an initial screen to determine the need for further clinical or genomic testing [74].

**Table 1 cancers-14-04025-t001:** Key characteristics of the selected articles.

Authors	Artificial Intelligence Method	Publication Year	Main Objectives	Title of Journal	Main Imaging Modality Used
Wang J. et al. [75]	Multi resolution technique, deep Siamese neural network.	2017	Detecting spinal metastases	Comput. Biol. Med.	MRI
Xiong X. et al. [76]	Radiomics using MRI machine learning techniques	2021	Differentiating spinal metastases subtypes and myeloma	Front. Oncol.	MRI
Filograna L. et al. [77]	Radiomics using MRI machine learning techniques	2019	Differentiating spinal metastases (feasibility)	Radiol. Med.	MRI
Zhong X. et al. [78]	Radiomics using MRI machine learning techniques	2020	Differentiating spinal metastases from osteoradionecrosis in nasopharyngeal carcinoma	BMC Med. Imaging	MRI
Chianca V. et al. [79]	Radiomics using MRI machine learning techniques	2021	Differentiating spinal metastases using several MRI scanners	Eur. J. Radiol.	MRI
Liu J. et al. [80]	Radiomics using MRI machine learning techniques	2022	Differentiating spinal metastases from multiple myeloma	Eur. Radiol.	MRI
Fan X. et al. [81]	Texture Analysis (radiomics-feature based) techniques	2020	Detect spinal metastases	Front Med. (Lausanne)	PET/CT
Naseri H. et al. [82]	Radiomics using CT machine learning techniques	2022	Detecting spinal metastases from unaffected bone	Scientific Reports	CT
Jin Z. et al. [83]	Radiomics using CT machine learning techniques	2021	Differentiating spinal metastases from non-aggressive/benign osseous lesions	Front Med. (Lausanne)	SPECT/CT
Yoda T. et al. [84]	Convolutional Neural Network (Deep learning)	2022	Differentiating spinal metastases and vertebral fractures from benign osteoporotic vertebral fractures	Spine (Phila Pa 1976)	MRI
Fan X. et al. [85]	Deep Learning (3D Convolutional Neural Network-based dilated convolutional U-Net algorithm)	2021	Detecting spinal metastases in lung cancer patients	Scientific Programming	Energy/Spectral CT images
Yao J. et al. [86]	Synthesis of CT and PET images, which provides lesion enhancement and aids computer detection	2017	Detecting spinal metastases	J. Med. Imaging (Bellingham)	CT
Mehta S. et al. [87]	Random forest classification technique	2019	Detecting spinal metastases (osteoblastic/sclerotic lesions)	Int. J. Comput. Assist. Radiol. Surg.	DEXA
Chang CY. et al. [88]	Convolutional Neural Network (Deep learning)	2022	Detecting spinal metastases (osteoblastic/sclerotic and treatment planning (Generating useful clinical parameters)	Skeletal Radiol.	CT
Roth H. et al. [89]	Convolutional Neural Network (Deep learning technique)-Random aggregation	2014	Detecting spinal metastases (osteoblastic/sclerotic)	Computational vision and biomechanics	CT
Wiese T. et al. [90]	Computer Aided Diagnosis using a watershed algorithm along with graph cut	2012	Detecting spinal metastases (osteoblastic/sclerotic)	Medical Imaging	CT
Burns J. et al. [91]	Fully-automated image analysis using a prototypical Computer Aided Diagnosis software	2013	Detecting thoracolumbar spinal metastases (osteoblastic/sclerotic)	Radiology	CT
Hammon M. et al. [92]	Automatic image analysis using Computer Aided Diagnosis software	2013	Detecting thoracolumbar spinal metastases (sclerotic/osteoblastic versus osteolytic)	Eur. Radiol.	CT
O’Connor S.D. et al. [93]	Automatic image analysis using Computer Aided Diagnosis software (preliminary)	2007	Detecting thoracolumbar spinal metastases (lytic lesion characterisation)	Radiology	CT
Hallinan J. et al. [94]	Deep Learning model/algorithm (convolutional neural network)	2022	Generating useful clinical parameters (classifying metastatic epidural disease and/or spinal cord compression)	Frontiers in Oncology	MRI
Arends S. et al. [95]	Deep Learning model/algorithm (convolutional neural network)	2020	Generating useful clinical parameters (planning radiation therapy for vertebral metastases)	Phys. Imaging Radiat. Oncol.	CT
Hille G. et al. [96]	Deep Learning model/algorithm (convolutional neural network)	2020	Generating useful clinical parameters (vertebral metastasis segmentation)	ArXiv	MRI
Lang N. et al. [97]	Deep learning techniques including radiomics	2019	Generating useful clinical parameters (classification of vertebral metastasis from lung cancer versus other malignancies)	Magn. Reson. Imaging	MRI/DCE
Wakabayashi K. et al. [98]	Three AI techniques/models used: Radiomic-features alone, clinical-features alone, and combined radiomics and clinical-feature algorithm	2021	Predicting prognosis (post-radiotherapy pain response for vertebral metastases)	Sci. Rep.	CT
Gui C. et al. [99]	Radiomics using CT and MRI machine learning techniques	2021	Predicting prognosis in spinal metastases (predicting the risk of spinal/vertebral compression fractures following stereotactic body radiation)	J. Neurosurg. Spine	CT & MRI
Yin P. et al. [100]	Radiomics using MRI (T2-weighted and post-contrast) machine learning techniques	2019	Differentiating spinal metastases (differentiation of lesions in the sacrum, e.g., chordoma, giant cell tumour, or metastastic lesions)	J. Magn. Reson. Imaging	MRI
Shi Y.J. et al. [101]	Radiomics using MRI machine learning techniques	2022	Predicting prognosis (response of lytic vertebral lesions to chemotherapy in patients with breast carcinoma)	Magn. Reson. Imaging	MRI
Ren M. et al. [102]	Radiomics using MRI machine learning techniques	2021	Generating useful clinical parameters (EGFR mutation prediction in lung cancer patients with thoracic vertebral metastases)	Med. Phys.	MRI
Fan Y. et al. [103]	Radiomics (subregional) using MRI machine learning techniques	2021	Generating useful clinical parameters (EGFR mutation prediction in lung cancer patients with thoracic vertebral metastases)	Phys. Med. Biol.	MRI
Cao R. et al. [104]	Radiomics (nomogram) using MRI machine learning techniques	2022	Generating useful clinical parameters/biomarkers (Prediction of EGFR mutations in exons 19/21 in lung cancer patients with thoracic vertebral metastases)	Academic Radiology	MRI

## 4. Discussion

Clinical Applications of AI in Spinal Metastases.

### 4.1. Detection of Spinal Metastases

Early detection and diagnosis of spinal metastases plays a key role in clinical practice. This will determine the stage of disease for the patient, and has the potential to alter the treatment regimen [105]. Metastatic spinal disease is associated with increased morbidity, and more than half of these patients will require radiotherapy or invasive intervention for complications, such as spinal cord or nerve root compression [106]. Hence, early diagnosis and treatment before permanent neurologic and functional deficits occur is essential for a favourable prognosis [107,108,109].

Manual detection of spinal metastasis through various imaging modalities is time consuming, tedious and often challenging with imaging features overlapping with many other pathologies. It is widely recognised that automated lesion detection could improve radiologist sensitivity for detecting osseous metastases, with computer-aided detection (CAD) software systems and artificial intelligence models proving to be as effective or even superior to manual radiologist detection [91,92,93]. Computer-assisted detection of spinal metastases was first studied on CT by O’Connor et al. [93] in 2007 for the detection of osteolytic spinal metastases. This paved the way for further studies using CAD, focusing on other subtypes of spinal metastasis such as osteoblastic or mixed type lesions [90], and other imaging modalities. Subsequently, with the recent advances in artificial intelligence in medical imaging [110,111,112], there were substantial improvements in the detection of spinal metastases with the aid of deep learning and convolutional neural networks. This has resulted in improvement in the accuracy of computer-assisted automated detection of spinal metastases across various imaging modalities with significant reduction in false positive and negative rates [75,85,88,89].

Wang J et al. [75] proposed an MRI-based detection of spinal metastases using a Siamese neural network model. This involved a multi-resolution technique to detect spinal metastasis with aggregation of neighbouring slices in MRI sequences to reduce false positive rates. Their methods were able to detect all spinal metastatic lesions from their datasets, with a relatively low false positive rate of 0.40 per case. Fan XJ et al. [85] developed a DC-U-Net model using energy/spectral CT imaging to improve detection of spinal metastases in patients with lung cancer. Their work utilised the fact that characteristic X-ray absorption by substances differs under various energy levels, which provides better detection, segmentation and differentiation of bone lesions. Their deep learning model showed performance close to that of an experienced physician, with detection rates of 66.03/81.41% by the deep learning model compared to 64.74%/77.56% by the professional doctor using high and low energy CT levels (140 kVp and 40 keV), respectively.

In another study Roth et al. [89] designed a two tier coarse-to-fine cascade framework incorporating existing computer-aided detection of sclerotic vertebral metastases on CT images with deep CNN classifications. Their proposed model with the help of CNN classifiers acts as an additional selective process to exclude difficult false positive results while preserving high sensitivities, reducing the false positives per volume (FP/vol.) from 4 to 1.2, 7 to 3, and from 12 to 9.5 when comparing sensitivity results of 60%, 70% and 80%, respectively, in their test set, with an AUC of 0.83.

AI has already shown its potential in other clinical fields for detection of pathology. For example, in breast cancer screening AI-assisted simulation software was developed by Raya-Povedano et al. [113] and showed workload reduction (in the form of manhours) of up to 70% for the triage of suspicious mammographic examinations compared to manual reading by radiologists, with non-inferior sensitivity of 84.1% and 16.7% lower recall rates. Although the application of AI in spinal metastasis detection is still in a preliminary stage, future clinical deployment would be important to reduce radiologists’ clinical workload, and reduce error rates with the AI algorithms acting as a second reader or safety net.

### 4.2. Differentiating Spinal Metastases from Other Pathological Conditions

Machine learning has been applied in several studies to help distinguish between spinal metastases and other pathology. This was first done by identifying key radiomics features in vertebral metastases [77], and incorporating this information with various machine learning models. For example, Liu et al. [80] and Xiong X et al. [76] utilised MRI-based radiomics to differentiate between spinal metastases and multiple myeloma, based on conventional T1-weighted (T1W) and fat-suppression T2-weighted (T2W) MR sequences. They incorporated the radiomics models using various machine learning algorithms such as Support-Vector Machine (SVM), Random Forest (RF), K-Nearest Neighbour (KNN), Naïve Bayes (NB) using 10-fold cross validation, Artificial Neural Networks (ANN) and Logistic Regression Classifier to predict the likelihood of spinal metastases. The radiomics model from Xiong X et al. used features from T2WI images, and achieved accuracy, sensitivity, and specificity of 81.5%, 87.9% and 79.0%, respectively, in their validation cohort. As for Liu et al., their model with 10-EPV (events per independent variable) showed good performance in distinguishing multiple myeloma from spinal metastases with an AUC of 0.85.

### 4.3. Pre-Treatment Evaluation

#### 4.3.1. Predicting Prognosis

Prediction of prognosis is a paradigm in oncological treatment. In patients with vertebral metastases, the ability to predict treatment response may help clinicians provide the most appropriate treatment with the best clinical outcome for the patient, avoid delayed transition to another treatment and prevent exposing patients to unnecessary treatment-related side effects. Shi YJ et al. [101] studied the value of MRI-based radiomics in predicting the treatment response of chemotherapy in a small group of breast cancer patients with vertebral metastases. Their radiomics model was effective in predicting progressive vs non-progressive disease with an area under the curve (AUC) of up to 0.91. This method could be extrapolated in future studies to predict the treatment response of spinal metastases and other primary tumours.

In addition to predicting treatment response, radiomics can also predict the effectiveness of treatment for symptom relief in patients with spinal metastasis. Back and neuropathic pain from spinal metastases are very common symptoms with many patients experiencing debilitating pain [114]. Radiotherapy can provide pain relief for spinal metastasis in certain situations. However, even palliative radiotherapy for vertebral metastases using relatively low doses of radiation, still has potential for adverse side-effects. Considering these risks, it will be helpful to identify those who have a high likelihood of pain relief from radiotherapy and direct other patients to alternate therapies. Wakabayashi K et al. [98] developed a radiomics model utilising feature subsets, random forests and recursive feature exclusion to predict pain response post-radiotherapy for vertebral metastases. Their study concluded that the model incorporating both clinical and radiomics features was the most effective in predicting response to pain following radiotherapy with an AUC of 0.85 and accuracy of 82.6%, and this was significantly improved (*p* = 0.044) when compared to the model only including clinical features with an AUC of 0.70 and accuracy of 65.2%.

#### 4.3.2. Identifying High Risk Vertebral Metastases Requiring Early Intervention

Applications of deep learning models goes beyond tumour detection and differentiation, and they have the ability to automatically generate meaningful parameters from MRI and other modalities. Hallinan et al. [94] developed a deep learning model for automated classification of metastatic epidural disease and/or spinal cord compression on MRI using the Bilsky classification. The model showed almost perfect agreement when compared to specialist readers on internal and external datasets with kappas of 0.92–0.98, *p* < 0.001 and 0.94–0.95, *p* < 0.001, respectively, for dichotomous Bilsky classification (low versus high grade). Accurate, reproducible classification of metastatic epidural spinal cord compression will enable clinicians to decide on initial radiotherapy versus surgical intervention [115].

Radiomic modelling and machine learning incorporating clinical features and radiomic features from pre-treatment imaging is also effective in predicting vertebral compression fractures 1 year following radiotherapy [99]. In a study performed by Gui et al., their radiomic machine learning model selected features from CT and MR images together with clinical features to predict vertebral compression fractures in patients with spinal metastases following stereotactic body radiation therapy. The combined radiomics/clinical model showed good performance with sensitivity of 84.4%, specificity of 80.0% and AUC of 0.88, exceeding the performance of clinical features alone (AUCs of 0.58 to 0.80). These studies show that meaningful parameters generated by machine learning models using radiomics and clinical information have the potential to triage urgent findings in spinal metastases, and identify those that require early intervention. Future work includes developing models that augment radiologist interpretation of spinal metastases, including automated disease stratification based on clinically useful classifications such as the Spinal Instability Neoplastic Score (SINS) [116]. SINS is useful to identify patients who require prompt surgical review and intervention.

#### 4.3.3. Pre-Treatment Planning and Monitoring

Segmentation refers to delineation or volume extraction of a lesion or organ based on image analysis. In clinical practice, manual or semi-manual segmentation techniques are being applied to provide further value to CT and MRI studies. However, these techniques are subjective, operator-dependent and very time-consuming which limits their adoption. Automatic segmentation of spinal metastases using deep learning models has been shown to be as accurate as expert annotations in both MRI and CT [88]. Hille G et. al. [96] showed that automated vertebral metastasis segmentation on MRI using deep convolutional neural networks (U-net like architecture) were almost as accurate as expert annotation. Their automated segmentation solution achieved a Dice–Sørensen coefficient (DSC) of up to 0.78 and mean sensitivity rates up to 78.9% on par with inter-reader variability DSC of 0.79. Potentially, these models will not only reduce the need for time-consuming manual segmentation of spinal metastases, but also support stereotactic body radiotherapy planning, and improve the performance [117,118] and treatment outcome of minimally invasive interventions for spinal metastasis such as radiofrequency ablation [95]. In respect to radiotherapy, precise automated tumour contours will improve treatment planning, reduce segmentation times and reduce the radiation dose to the surrounding organs at risk, including the spinal cord. In recent years, various image segmentation techniques have been proposed, resulting in more accurate and efficient image segmentation for clinical diagnosis and treatment [119,120,121,122].

The efficiency of AI algorithms to perform repetitive tasks, such as segmentation have already been shown to outperform manual approaches in various clinical studies. A study by Winkel D. et al. [123,124] showed that their fully automated liver segmentation algorithm using deep learning was able to achieve a mean processing time of 9.94 s, at least 20 times faster than manual segmentation with excellent agreement between the two approaches (intraclass correlation coefficient of 0.996). In a polycystic liver and kidney disease series of CT images, an artificial intelligence model segmented the liver parenchyma at 8333 slices/hour, compared to labour intensive manual segmentation by an expert clinician, which did not surpass 16 slices/hour, with a DSC of 0.96 [125]. While the current AI models and applications on spinal metastasis segmentation are still in a preliminary stage, their use could help reduce clinical workload and improve productivity when the technology is translated into clinical care.

Artificial intelligence applications could also help to improve measurement of tumour burden for assessment of treatment response and monitoring of tumour progression. This can be achieved by obtaining lesion and/or tumour volumetry through AI-assisted segmentation. For example, Goehler et al. [126] developed a deep learning approach to estimate overall tumour burden for neuroendocrine neoplasia on MR images, achieving concordance with manual clinician assessment in 91% with sensitivity of 85.0%, specificity of 92.0% and DSC of up to 0.81. Estimating tumour burden and volume of disease in spinal metastases is difficult due to the shape of the vertebrae and presence of multiple vertebral levels. Tumour burden of spinal metastases has been shown to predict prognosis [106,127,128,129] and treatment efficiency [130]. With the help of AI, volumetric evaluation of spinal metastases and disease burden can be more efficient, clinically feasible and could aid clinical management.

### 4.4. Radiogenomics and Phenotyping

Radiogenomics, the combination of “*Radiomics*” and “*Genomics*”, refers to the use of imaging features or surrogates to determine genomic signatures and advanced biomarkers in tumours. These biomarkers can then be used for clinical management decisions, including prognostic, diagnostic and predictive precision of tumour subtypes [131]. The workflow of a radiogenomics study can be commonly classified into five different stages (Figure 5): (1) image acquisition and pre-processing, (2) feature extraction and selection from both the medical imaging and genotype, (3) association of radiomics and genomics features, (4) data analysis using machine learning models and (5) final radiogenomics outcome model [132].

In the case of spinal metastasis, several radiogenomics models in patients with primary lung malignancy [102,103,104] have the ability to predict the presence or absence of the EGFR mutation through the analysis of MR images of the spine. For instance, the multiparametric MRI-based radiomics model developed by Ren M et al. [102] combining both radiomics and clinical features achieved AUC of up to 0.89, sensitivity of 83.9% and specificity of 79.3% in differentiating the EGFR mutation versus EGFR wild-type patients in their training cohort, whereas the radiomics nomogram developed by Cao R et al. [104] was able to obtain good prediction performance of EGFR mutation in thoracic spinal metastases from MRI-based images with an AUC of up to 0.90. These non-invasive, quantitative and convenient methods to predict the EGFR mutation in spinal metastases may help guide individualised treatment in lung adenocarcinoma.

To date, there are no other published studies on radiomics models for identifying other clinically important genotypes in spinal oncology imaging, such as KRAS, BRAF and ALK [133], which have been shown to have diagnostic and prognostic significance. However, there are many studies in the literature for other organs, including breast oncology [134] imaging (e.g., miRNAs expression, gene expression and Ki67 proliferation index), and for brain imaging (Appendix A), including biomarkers such as isocitrate dehydrogenase (IDH) [135], chromosome arms 1p/19q-codeletion [136,137,138] and methylguanine-DNA methyltransferase status (MGMT) [139,140,141,142] as prognostic markers for glioma [143,144,145]. Future work will include the study of these other clinically important genotypes to aid in deciding treatment for patients with spinal metastases or other primary tumours, supporting the new era of precision medicine [146,147].

Spinal metastasis from an unknown primary is a common clinical dilemma seen in up to 30% of patients upon initial presentation of spinal metastatic lesions [148,149,150]. While conventional MRI can provide accurate detection of metastases in the vertebrae, cancers from different primary tumours can appear similar and may be difficult to distinguish [151,152]. These patients often require further PET/CT imaging for diagnosis of the primary cancer and whole-body staging before treatment can take place. In rare circumstances even with further imaging the primary cancer cannot be identified, and invasive biopsy is required to ascertain the likely primary tumour site and treatment options. Lang N et al. [97] developed radiomics and DL models to differentiate spinal metastases originating from lung and other cancers using dynamic contrast enhanced sequences (DCE) from a spinal MRI database. Their deep learning models demonstrated that the DCE kinetic measurement of the washout slope from a hotspot of the spinal metastatic lesion was the most accurate parameter to aid diagnosis of primary lung cancer from other tumours. Their deep learning convolutional long short-term memory network using the whole database of DCE images had accuracy of up to 81.0%. Being able to predict the primary malignancy in a patient with spinal metastases may help guide clinicians on the most relevant investigations, thereby reducing unnecessary evaluations, medical costs and minimising the need for invasive spinal biopsy.

### 4.5. Post-Treatment Evaluation

#### 4.5.1. Residual/Recurrent Tumour versus Post-Treatment Changes

A common and challenging clinical scenario is differentiating spinal metastases from post-treatment changes. As the clinical treatment between these two entities are vastly different, accurate diagnosis is important to prevent unnecessary invasive biopsy and/or chemoradiotherapy. An example is osteoradionecrosis in the cervical spine, which is a complication following radiotherapy in patients with nasopharyngeal carcinoma [153]. Several studies have shown that osteoradionecrosis can mimic vertebral metastases as both can present with soft tissue masses and abnormal enhancement on MRI [154,155]. Machine learning radiomics has been shown to be useful as a non-invasive visual diagnostic tool to differentiate between the two entities. Zhong et al. [78] created an MRI-based radiomics nomogram that was shown to be clinically useful in discriminating between cervical spine osteoradionecrosis and metastases, with an AUC of 0.73 on the training set and 0.72 in the validation set.

Vertebral compression fractures following radiotherapy of spinal metastases occur in up to 9.4–12% [156,157] of cases, and are the most common and serious side effect of stereotactic body radiotherapy. It is often difficult to assess whether a compression fracture is related to tumour progression or radiation induced fibrosis or necrosis [158,159]. To date, there are no published radiomics studies differentiating vertebral compression fractures as a result of radiotherapy versus those related to tumour progression. However, Yoda et al. [84] developed a CNN model for differentiating osteoporotic from metastatic vertebral compression fractures and showed high accuracy using MRI-based T1WI features, achieving AUC of 0.98, 96.4% accuracy, 98.1% sensitivity and 94.9% specificity, which was statistically equal or superior to that of spine surgeons. A similar approach may be applied for differentiating vertebral compression fractures following radiotherapy from those related to tumour progression, and this could leverage transfer learning given the similar topics [73].

#### 4.5.2. Pseudo-Progression

Pseudo-progression is a post-treatment phenomenon involving an increase in the target tumour volume (usually without any worsening symptoms), which then demonstrates interval stability or reduction in volume on repeat imaging. It occurs in approximately 14 to 18% of those with vertebral metastases treated with stereotactic body radiotherapy [160,161]. The differentiation of pseudo-progression from true progression is challenging on imaging even with many studies suggesting some differentiating factors [161,162], such as location of involvement, e.g., purely vertebral body involvement with pseudo compared to involvement of the epidural space with true progression. Artificial intelligence has already shown utility in aiding the differentiation of pseudo from true progression in brain imaging. Kim et al. [163] and Jang et al. [164] demonstrated good performance of their radiomics models in differentiating true progression of glioblastoma following surgical resection and radio-chemotherapy versus pseudo-progression post-treatment. These studies demonstrate the potential of AI in differentiating these two entities in spinal imaging, and future work may contribute to earlier suspicion of true progression allowing for close imaging follow-up, and earlier diagnosis [165].

## 5. Conclusions

Artificial intelligence, including machine learning technologies, has achieved good performance in spine oncology. These techniques have immense potential to aid clinicians in the management of spinal metastases including treatment selection, enhancement of workflow efficiency and reduction in complications and adverse events. However, the majority of the studies are preliminary, retrospective or based at a single-centre with small sample sizes. As a result, the models developed in these studies are subjected to limited generalisability with significant heterogeneity in results when applied to external datasets. This can result in reduced reproducibility of the results and may impede the development of AI models that can be translated successfully into clinical use. Further research, especially randomised controlled trials or large multi-centre studies, is required to validate these applications and facilitate their integration into routine clinical practice.

## Figures and Tables

**Figure 1 cancers-14-04025-f001:**
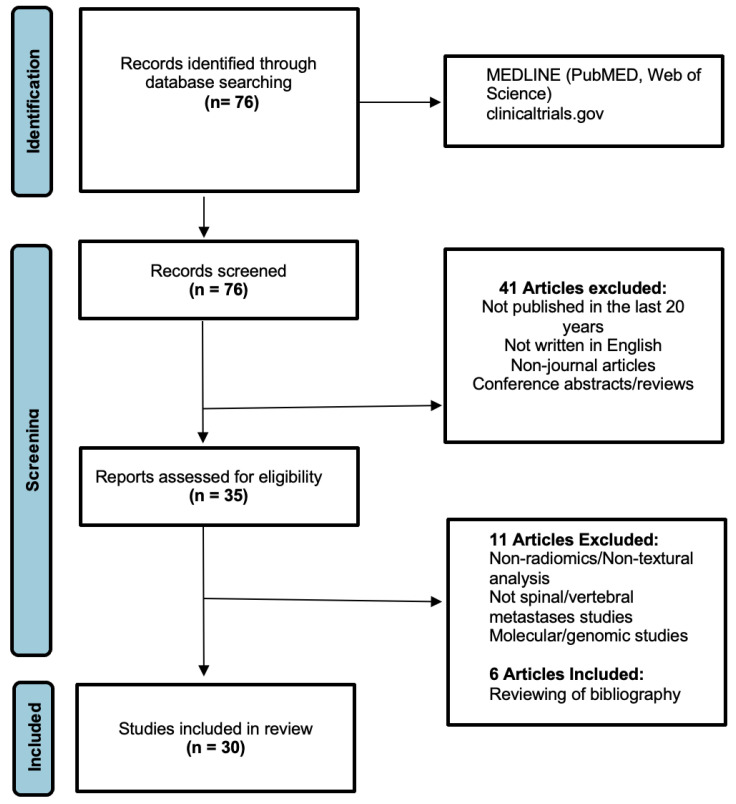
PRISMA flowchart for the literature search (this is adapted from the PRISMA group, 2020), which describes the selection of relevant articles.

**Figure 2 cancers-14-04025-f002:**
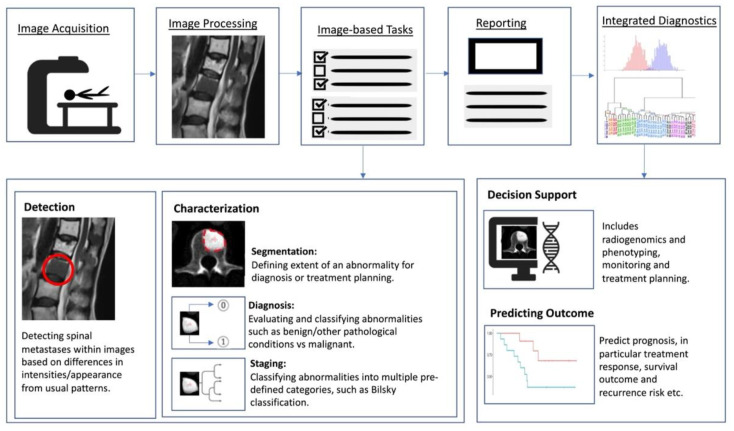
Schematic outline showing where AI implementation can optimise the radiology workflow. The workflow comprises the following steps: image acquisition, image processing, image-based tasks, reporting, and integrated diagnostics. AI can add value to the image-based clinical tasks, including the detection of abnormalities; characterisation of objects in images using segmentation, diagnosis and staging; and integrated diagnostics including decision support for treatment planning and prognosis prediction.

**Figure 3 cancers-14-04025-f003:**
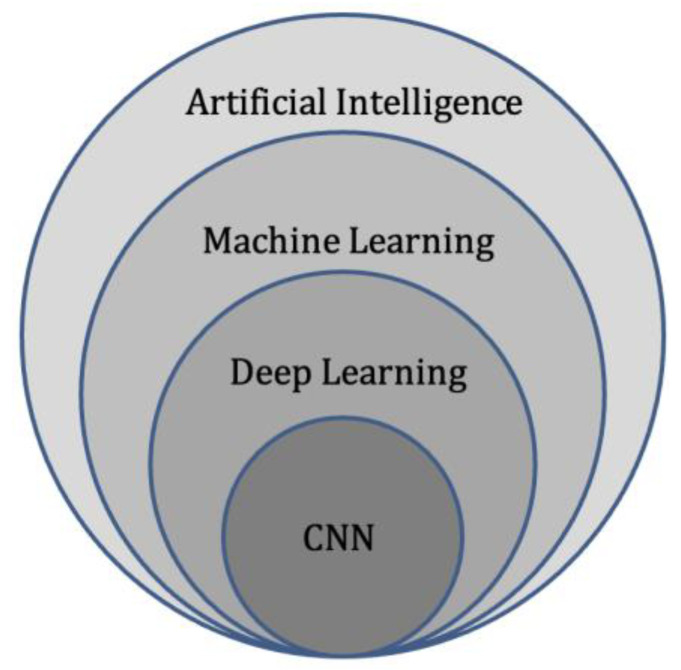
Diagram of artificial intelligence hierarchy. Machine learning lies within the field of artificial intelligence and is an area of study that enables computers to learn without explicit knowledge or programming. Within machine learning, deep learning is another area of study that enables computation of neural networks involving multiple layers. Finally, convolutional neural networks (CNN) are an important subset of deep learning, commonly applied to analyse medical images.

**Figure 4 cancers-14-04025-f004:**
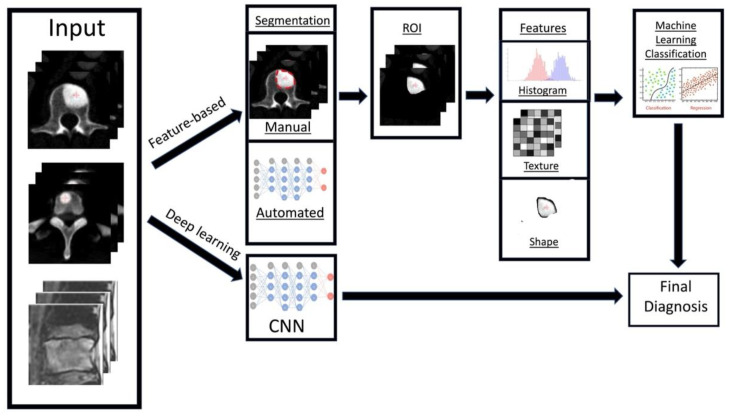
Diagram showing the general framework and main steps for radiomics, namely data selection (input), medical imaging evaluation and segmentation, feature extraction in the regions of interest (ROIs), exploratory analysis and modelling.

**Figure 5 cancers-14-04025-f005:**
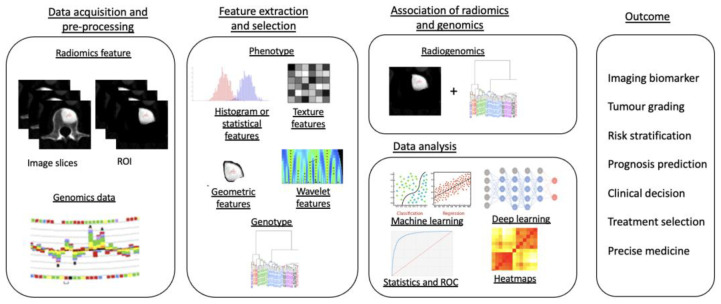
Diagram showing a five-stage radiogenomics pipeline including data acquisition (radiological imaging) and pre-processing, feature extraction and selection, subsequent association of radiomics techniques and genomics, analysis of data and model development and, finally, radiogenomics outcomes.

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
