# Peer review of "Application of Artificial Intelligence Methods for Imaging of Spinal Metastasis"

_cancers, 2022, doi:10.3390/cancers14164025_

Round 1
Reviewer 1 Report
Comment 1.
The authors should explain a detailed example regarding Radiogenomics in their manuscript.
For instance, add a new Figure and paragraph explaining a discovery of isocitrate dehydrogenase (IDH) mutation, co-deletion of chromosome arms 1p/19q, and methylguanine-DNA methyltransferase promoter methylation status (MGMT) in glioblastoma patients by using Radiogenomics approaches into their revised manuscript.
Author Response
Reviewer 1:
The authors should explain a detailed example regarding Radiogenomics in their manuscript. For instance, add a new Figure and paragraph explaining a discovery of isocitrate dehydrogenase (IDH) mutation, co-deletion of chromosome arms 1p/19q, and methylguanine-DNA methyltransferase promoter methylation status (MGMT) in glioblastoma patients by using Radiogenomics approaches into their revised manuscript.
R1.1: Thank you for the suggestion. We have changed the text and added a supplementary figure 1, which shows the framework of how Radiogenomics are derived for predicting IDH-mutation, 1p/19q mutation and MGMT-methylation status.
Reviewer 2 Report
-The authors have discussed the Machine learning and AI which have become more popular in oncological imaging. This article reviews and summarizes the evidence supporting AI applications in the diagnosis, classification, and management of spinal metastases, as well as their potential therapeutic integration. My comments are below:
-The authors in this manuscript have a conducted a very well designed study and overall the paper is very informative and the discussion was extensive.
-Also, the author mentioned that " In spine cancer, AI has shown good performance and has the potential to help physicians improve labor efficiency and reduce adverse effects." How about other cancers what is the progress so far.
- Image processing, diagnosis, decision support, therapeutic help, and predictive outcomes are clinical AI uses. If adding graphical abstract that help understanding the uses in clinical setting that will be good addition to the article.
Finally, I suggest to accept the manuscript after rearranging the figures after the text of first mentions. Also, There are some very long paragraph and I suggest to break them up.
Author Response
Reviewer 2:
-Also, the author mentioned that " In spine cancer, AI has shown good performance and has the potential to help physicians improve labor efficiency and reduce adverse effects." How about other cancers what is the progress so far.
R2.1: Thank you for the suggestion. This subject was briefly mentioned in the discussion using several examples, e.g., AI for differentiation of pseudo from true progression in brain imaging, radiomics models for breast and brain imaging, and AI estimation of tumour burden in neuroendocrine neoplasia on MR images. We have included a more comprehensive sentence on AI applications for other cancers in the introduction.
- Image processing, diagnosis, decision support, therapeutic help, and predictive outcomes are clinical AI uses. If adding graphical abstract that help understanding the uses in clinical setting that will be good addition to the article.
R2.2: We agree this would be a useful addition to the manuscript. We have added figure 2 and a paragraph to illustrate the clinical application of AI for spinal oncology imaging.
- Finally, I suggest to accept the manuscript after rearranging the figures after the text of first mentions. Also, There are some very long paragraph and I suggest to break them up.
R2.3: We have rearranged the figures and broke up some long paragraphs. Thank you for reviewing our manuscript.